# Obesity Is Associated with Immunometabolic Changes in Adipose Tissue That May Drive Treatment Resistance in Breast Cancer: Immune-Metabolic Reprogramming and Novel Therapeutic Strategies

**DOI:** 10.3390/cancers15092440

**Published:** 2023-04-24

**Authors:** Constantinos Savva, Ellen Copson, Peter W. M. Johnson, Ramsey I. Cutress, Stephen A. Beers

**Affiliations:** 1Antibody and Vaccine Group, Centre for Cancer Immunology, School of Cancer Sciences, Faculty of Medicine, University of Southampton, Southampton SO16 6YD, UK; 2CRUK Southampton Centre, School of Cancer Sciences, Faculty of Medicine, University of Southampton, Southampton SO16 6YD, UK; 3Southampton Experimental Cancer Medicine Centre, School of Cancer Sciences, Faculty of Medicine, University of Southampton, Southampton SO16 6YD, UK; 4NIHR Southampton Biomedical Research Centre, University Hospital Southampton NHS Foundation Trust, Southampton SO16 6YD, UK

**Keywords:** breast cancer, obesity, inflammation, metabolism, treatment resistance

## Abstract

**Simple Summary:**

Obesity is associated with metabolic changes in the immunological tumour infiltrate in breast cancer. The crosstalk between adipocytes, macrophages and proinflammatory cytokines may promote immunometabolic dysregulation and immunosuppressive phenotypes in breast tumours which may be correlated to treatment resistance. In this review, we provide a comprehensive overview of emerging evidence describing the association between obesity and immunometabolic dysfunction in breast cancer and discuss novel therapeutic strategies to overcome immunosuppressive phenotypes.

**Abstract:**

White adipose tissue (WAT) represents an endocrinologically and immunologically active tissue whose primary role is energy storage and homeostasis. Breast WAT is involved in the secretion of hormones and proinflammatory molecules that are associated with breast cancer development and progression. The role of adiposity and systemic inflammation in immune responses and resistance to anti-cancer treatment in breast cancer (BC) patients is still not clear. Metformin has demonstrated antitumorigenic properties both in pre-clinical and clinical studies. Nevertheless, its immunomodulating properties in BC are largely unknown. This review aims to evaluate the emerging evidence on the crosstalk between adiposity and the immune-tumour microenvironment in BC, its progression and treatment resistance, and the immunometabolic role of metformin in BC. Adiposity, and by extension subclinical inflammation, are associated with metabolic dysfunction and changes in the immune-tumour microenvironment in BC. In oestrogen receptor positive (ER+) breast tumours, it is proposed that these changes are mediated via a paracrine interaction between macrophages and preadipocytes, leading to elevated aromatase expression and secretion of pro-inflammatory cytokines and adipokines in the breast tissue in patients who are obese or overweight. In HER2+ breast tumours, WAT inflammation has been shown to be associated with resistance to trastuzumab mediated via MAPK or PI3K pathways. Furthermore, adipose tissue in patients with obesity is associated with upregulation of immune checkpoints on T-cells that is partially mediated via immunomodulatory effects of leptin and has been paradoxically associated with improved responses to immunotherapy in several cancers. Metformin may play a role in the metabolic reprogramming of tumour-infiltrating immune cells that are dysregulated by systemic inflammation. In conclusion, evidence suggests that body composition and metabolic status are associated with patient outcomes. To optimise patient stratification and personalisation of treatment, prospective studies are required to evaluate the role of body composition and metabolic parameters in metabolic immune reprogramming with and without immunotherapy in patients with BC.

## 1. Introduction

BC is the most common type of malignancy among women worldwide [1]. The worldwide average age-standardised incidence and mortality rates were estimated to be 47.8 and 13.6 per 100,000 persons per year, respectively, in 2020 [1]. Therefore, BC is a major public health burden that needs to be addressed not only through primary prevention and early diagnosis but also via effective personalised treatments [2].

The increasing prevalence of obesity (body mass index [BMI] ≥ 25 kg/m^2^) has also become an important public health concern [3,4]. The worldwide prevalence of obesity increased dramatically by 27.5% for adults between 1980 and 2013 [3]. Overweight and obesity not only increase the risk of developing BC but also are associated with worse survival compared to patients with normal weight [4,5]. BMI is used as a surrogate of abnormal or excessive fat accumulation; however, it is not a reliable measure of body adiposity as BMI does not consider the percentages of lean and fat tissue mass [6]. Furthermore, the same BMI may correspond to different lean and body fat mass that change with sex, ethnicity and age [4]. Therefore, further research is required to understand how body composition and BC interact.

Western lifestyle is associated with low-grade inflammation and chronic metabolic inflammatory diseases linked to decreased life expectancy [7]. Risk factors such as Western diet, reduced physical activity, environmental or socioeconomic factors such as smoking, income, education and occupation, can result in chronic systemic inflammation that activates local immune cells, including macrophages, either directly or indirectly mediated via obesity [7,8,9]. In the context of nutrient excess, adipocytes increase in number and size, which results in adipocyte disruption and cell death [10]. The innate immune cells, such as macrophages, respond to environmental danger signals that are released by the necrotic adipocytes that trigger chronic low-grade systematic inflammation via metabolic reprogramming [11]. During inflammation, there is an interplay between macrophages, which act as antigen-presenting cells, and adaptive immunity cells such as T-cells, by shaping T-cell responses and dysregulating the immunometabolic homeostasis towards a proinflammatory environment [12].

Metformin is a dimethyl biguanide antidiabetic drug with pleiotropic properties including effects that involve tumour metabolism and inflammatory pathways [13]. Metformin was associated with an improvement in anthropometric parameters such as waist circumference and waist-to-hip ratio, as well as a reduction in markers of systemic inflammation in premenopausal women with high BMI without history of BC compared to the control arm, suggesting that it may modulate obesity-induced inflammation [14]. Furthermore, both clinical and pre-clinical studies have shown that metformin has a direct anti-tumour effect and modulates the tumour microenvironment via the enhancement of anti-tumour immune responses [13].

In this review, we evaluate the emerging evidence on the crosstalk between adiposity and the tumour-immune microenvironment in BC progression and treatment resistance as well as the immunometabolic effects of metformin.

## 2. Immunometabolic Changes in Adipose Tissue of Patients with Obesity

Under normal physiological conditions, in adipose tissue of individuals with healthy BMI, there is homeostasis between anti-inflammatory and proinflammatory molecules that maintains adipose tissue functions [15]. Adipose tissue is classified into white and brown adipose tissue (WAT and BAT, respectively) which differ functionally and morphologically. WAT, distributed subcutaneously and viscerally, constitutes 20% of the body weight and 80% of total adipose tissue of a normal adult [16]. WAT is the largest store of energy, whilst BAT plays a key role in thermogenesis [16]. Under conditions of nutritional excess and during development of obesity, adipocytes undergo structural alterations, such as adipocyte hypertrophy [17]. Then adipocytes become dysfunctional, undergo cell death and secrete cytokines that contribute to adipose tissue inflammation and recruitment of pre-adipocytes, leading to adipose tissue hyperplasia [18,19]. Apart from structural changes, adipose tissue also undergoes functional changes such as mitochondrial dysfunction and endoplasmic reticulum stress [20]. In obesity, there is a significant reduction in mitochondrial gene expression leading to downregulation of mitochondrial biogenesis in subcutaneous tissue associated with insulin resistance and inflammation in obese monozygotic twins compared with their leaner co-twins [21]. Furthermore, free fatty acid-mediated generation of reactive oxygen species is correlated with endoplasmic reticulum stress and upregulation of pro-inflammatory gene signatures in adipose tissue [22,23]. Overall, these changes in the adipose tissue result in the activation of proinflammatory signalling pathways, leading to chronic low-grade adipose tissue inflammation mediated by macrophage infiltration, neovascularisation and increase in extracellular matrix [24,25,26,27] (Table 1). BC arises in an adipose rich environment and therefore its tumour microenvironment could be impacted by these factors.

## 3. Innate Immunity in Adipose Tissue of Patients with Obesity

During weight gain, adipocytes increase the storage of lipids, resulting in structural changes, such as adipocyte hypertrophy, and adipocyte death. The mechanism of adipocyte death is still not clear, although it has been attributed to either inflammatory programmed cell death (pyroptosis) or necrosis [38,39]. Adipose tissue macrophages (ATMs) scavenge the debris of the necrotic adipocytes, which in turn activate ATMs via the initiation of inflammatory signalling pathways [40,41,42,43]. ATMs that are metabolically activated by fatty acids or inflammatory mediators that are released from necrotic adipocytes are recruited or proliferate in situ, and encircle adipocytes forming crown-like structures (CLSs) (Figure 1) [44,45].

Obesity was reported to modulate the phenotype of ATMs from an M2- to M1-like phenotype in mice that received a high fat diet [46]. ATMs in obese adipose tissue express M2-like markers such as CD163 or CD206, and activate Fcγ receptors such as CD16, as well as a range of M1-like markers, including CD11c, which is involved in T-cell activation in adipose tissue [45,47]. This CD11c+ CD163+ subset of ATMs is associated with high BMI and accumulates in the adipose tissue of obese subjects [48]. CD11c+ CD206+ ATMs confer pro-inflammatory properties that correlate with increased presence of CLS and insulin resistance in obese individuals [49]. This suggests that ATMs and consequently CLSs are a diverse immune cell population defined by concurrent expression of biomarkers for M1- and M2-like macrophages that is dependent on the presence of adipose tissue in obese people and most likely driven by metabolic dysfunction [40,49].

## 4. Adaptive Immunity in Adipose Tissue of Patients with Obesity

During the development of obesity, the composition of adaptive immune cells resident in adipose tissue changes, with an increase in the CD8+ to CD4+ T-cell ratio, but a reduction in the number of regulatory T-cells (Tregs) within the adipose tissue [50]. Both CD4+ and CD8+ T-cells play a vital role in the recruitment and activation of ATM via secretion of cytokines such as IFN-γ [29]. Flow-cytometric and immunohistochemical analyses demonstrated higher numbers of CD8+ effector T-cells and lower numbers of CD4+ helper T-cells in obese murine epididymal adipose tissue compared to lean mice on a normal diet [29]. In addition, CD8+ T-cells were found within CLS in obese epididymal adipose tissue whereas there was no association was found between CD4+ T-cells and CLS [29]. A time course evaluation of immune cells in adipose tissue in C57BL/6 mice during a high-fat diet showed that CD8+ T-cell infiltration preceded the recruitment of macrophages [29]. In contrast, the number of CD4+CD8- helper T-cells and CD4+CD25+FoxP3+ Tregs decreased, suggesting that CD8+ T-cell infiltration is a crucial event during inflammation in adipose tissue [29]. This was further validated by depleting CD8+ T-cells in C57BL/6 mice using anti-CD8 antibody, which resulted in reduction of M1-like macrophages and CLSs without affecting M2-like macrophages [29]. In addition, high-fat diet did not increase levels of IL-6 and TNF-α mRNA in CD8-deficient mice, whereas adoptive transfer of CD8+ T-cells into CD8-deficient mice increased M1-like macrophage infiltration [29].

These findings suggest that Tregs maintain immune homeostasis by suppressing inflammation induced by pro-inflammatory macrophages in adipose tissue under physiological conditions. CD8+ T-cell infiltration is required for adipose tissue inflammation in obesity as it precedes macrophage accumulation in adipose tissue and plays a vital role in macrophage polarisation and infiltration. Obesity-induced metabolic dysregulation may interfere in the interplay between macrophages and T-cell immune populations [29]. Differences in the immunophenotype of ATMs between non-obese and obese subjects may be attributed to their different immunometabolic functions influenced by metabolic stress and chronic inflammation, which is promoted by enlarged or necrotic adipocytes.

## 5. Adipose Tissue Macrophages and Breast Cancer

CLSs are correlated with a proinflammatory environment and represent an index of WAT inflammation and metabolic dysregulation such as dyslipidemia, increased glucose and glycated hemoglobin (HbA1) levels [44,45,51,52]. Observational studies in patients with early breast cancer demonstrated that chronic systemic inflammation, as defined by elevated serum proinflammatory cytokines, is associated with CLSs, especially in individuals with obesity or who are overweight [51,53,54]. Previous reports have demonstrated an inconsistency in survival, with three out of five studies reporting improved outcomes (Table 2), which can be partly explained by biological and methodological heterogeneity [45,51,55,56]. Our recent study showed that the presence of CLSs expressing the inhibitory FcγRIIB (CD32B) at the tumour border was associated with worse clinical outcomes in patients with HER2+ breast cancer treated with trastuzumab compared to trastuzumab-naïve patients [45]. The underlying biological mechanism that links the presence of CD32B+ CLS and resistance to trastuzumab is currently unclear and further investigation is required. We hypothesise that this observation may be attributed either to a direct effect on the immune system or could be an index of inflammation within the tumour microenvironment. An improved trastuzumab-dependent cell-mediated cytotoxicity was reported in a mouse xenograft breast tumour model that was not expressing CD32B, whereas mice deficient in activating FcγRs were unable to supress tumour growth [57]. We hypothesise that the spatial distribution of CLSs along with the promotion of chronic inflammation may be associated with resistance to anti-HER2 treatment [45].

Griner et al. investigated the role of adipocytes in trastuzumab resistance in HER2+ BT474 and SKBR-3 cell lines. Culture of these cell lines with conditioned media from in vitro differentiated adipocytes was associated with improved viability and AKT phosphorylation in the trastuzumab-treated HER2+ cell lines compared to the controls [60]. Pharmacological blockade of PI3K via the PI3K inhibitor LY294002 or transfection with an inactive AKT1 kinase mutant reversed the resistance to trastuzumab that was induced by the conditioned media [60]. It was also reported that leptin enhances the overexpression of HER2 receptor and cell proliferation of HER2+ breast cancer cell lines, which results in resistance to tamoxifen [61,62]. Leptin-induced overexpression of the HER2 receptor is mediated via the activation of the RAS-dependent MAPK pathway, which phosphorylates both epidermal growth factor receptor and Janus-activated kinase 2 [61,62]. Stimulation of the MCF7 cell line with leptin was associated with HER2 phosphorylation on Tyr1248, which led to the activation of proliferation and survival pathways [63]. Leptin-dependent activation of the HER2 receptor could be explained by the co-localisation of leptin and HER2 receptors in HER2+ cell lines and human breast tumours [63].

In ER+ BC, aromatase expression and activity are associated with both high BMI and WAT inflammation [64]. Interestingly, WAT inflammation, as defined by the presence of CLSs, is associated with raised aromatase expression and activity in women with normal BMI [54,65]. Furthermore, adipocyte size and markers of subclinical systemic inflammation are strongly associated with increased levels of aromatase in postmenopausal women [64]. In addition, menopause is associated with a reduction in oestrogen levels, which is linked to the development of obesity [66]. Treatment with 17β-oestradiol protected ovariectomised mice against high fat diet induced weight gain and was associated with a reduction in aromatase expression, WAT inflammation and the associated proinflammatory mediators in the mammary glands, which are mediated via oestrogen receptor-α [66]. Hence, administration of oestrogen in obese mice may modulate WAT inflammation either through weight loss or due to its potential anti-inflammatory properties [66].

These findings suggest that hyperadiposity can induce WAT inflammation and metabolic dysregulation. In ER+ breast tumours, this is potentially mediated via a paracrine interaction between macrophages and preadipocytes, leading to elevated aromatase expression and secretion of pro-inflammatory adipokines in the breast adipose tissue in patients with high BMI [44]. In contrast, in HER2+ breast tumours, WAT inflammation can induce trastuzumab resistance via activation of MAPK or PI3K pathways [60].

## 6. BMI and Clinical Outcomes in Different Breast Cancer Subtypes and Responses to Treatment

Previous reports showed that obesity is associated with immunosuppressive changes in the tumour microenvironment that can be reversed with the use of immune checkpoint inhibitors (ICPIs) [67]. Nonetheless, there is conflicting evidence about the role of obesity in therapeutic responses in BC that can be explained by the biological heterogeneity of breast tumours.

## 7. HER2+ Breast Cancer

Clinical studies in patients with HER2+ breast tumours demonstrated a discrepancy between BMI and clinical outcomes, including pathological response and overall survival, which may be explained by tumour heterogeneity in hormonal receptor profile, inclusion of overweight patients and intensity of anti-HER2-directed treatment within both the early and metastatic settings [68,69].

Subclinical inflammation has also been correlated with therapeutic responses in patients with HER2+ BC. A cross-sectional study that included 175 patients with luminal B HER2-positive tumours showed that these patients had a 2 times higher risk of having a waist circumference of ≥80 cm and a 3 times higher CRP compared to luminal A patients [70]. In a cohort of 66 patients with metastatic HER2+ BC treated with trastuzumab alone or trastuzumab and chemotherapy, patients with elevated markers of systemic inflammation prior to treatment demonstrated worse progression-free survival and shorter OS [71]. Pre-clinical studies showed that *PTEN* deletion-induced resistance in HER2-amplified BC cell lines was mediated by a significant increase in the secretion of proinflammatory cytokines including IL6 [72]. IL-6 induced a positive feedback loop that was dependent on NF-kB signalling and resulted in the generation of a cancer stem-like cell (CSC) population. Pharmacological inhibition of IL6 receptor signalling, alone or in combination with trastuzumab, blocked this inflammatory feedback loop and led to a decrease in the CSC population, tumour growth and metastasis in mouse xenografts [72]. Shou Liu et al. demonstrated that HER2+ overexpression promoted tumour-derived secretion of IL1a and IL6 which, in turn, induced the NF-kB and STAT3 pathways and led to the expansion of breast CSCs [73]. Furthermore, in human breast tumour samples, there was strong evidence of an association between high IL1a/IL6 expression and the CSC-positive phenotype. In addition, in IL1a knock-out mice, there was inhibition of HER2-induced tumorigenesis and reduction in inflammatory cytokine secretion. Chemical inhibition of IL1a signalling reduced the CSC population and improved chemotherapeutic responses both in vitro and in vivo [73]. These demonstrate that inflammation mediates BC progression and metastatic spread and treatment resistance in pre-clinical models and clinical studies.

## 8. Oestrogen Receptor Negative Breast Cancer

The role of adiposity and inflammation was also demonstrated in ER−BC. A systematic review and meta-analysis that included 13 observational studies of patients with TNBC with baseline BMI measurements, showed that BMI ≥ 25 was associated with worse disease-free and overall survival compared to patients with healthy BMI [74]. Although these studies were characterised by low risk of bias, there was a statistically significant heterogeneity among the studies, which limits the interpretation of the results [74]. In a study of 1779 patients with primary invasive BC, patients with triple-negative disease had a 3-fold risk of being overweight and of having raised CRP compared to luminal A subjects [70].

The associations between BMI and gene expression of both tumour and adjacent tissue were investigated in 519 postmenopausal women from the Nurse’s Health Study [75]. In ER− tumours and tumour-adjacent tissues, high BMI was associated with enhanced inflammation pathways with increased expression of genes associated with IFN-α and INF-γ response and activated mTORC1 complex. Tumour-adjacent tissues in ER− disease displayed activated inflammation pathways including IL-6 and IFN gamma with increasing BMI [75]. Recent studies proposed potential mechanisms underlying the association between obesity and TNBC. These include the activation of Akt/mTOR signalling pathway by insulin, which is elevated in patients with obesity-induced insulin resistance [76]. Activation of the Akt/mTOR pathway is associated with aggressive molecular and glycolytic phenotypes that promote tumour growth in TNBC [77,78]. Secondly, obesity-mediated inflammation has been associated with activation-signalling pathways that are involved in tumour invasion and metastasis in TNBC [79]. Thirdly, this chronic inflammatory environment was reported to be correlated with reduced tumour-infiltrating immune cells both in a 4T1 TNBC model and in human triple negative tumours [80]. Hence, obesity is positively associated with inflammatory and aggressive molecular phenotypes in patients with TNBC.

## 9. Oestrogen Receptor Positive Breast Cancer

Gene set enrichment analysis (GSEA) of ER+ tumours from the Nurse’s Health Study showed that high BMI was significantly correlated with upregulated cellular proliferation pathways in the primary tumours and epithelial mesenchymal transition and inflammatory pathways in the tumour-adjacent tissues [75]. In a study of 137 patients with ER+ BC, obesity was associated with shorter overall and progression-free survival compared to patients without obesity [81]. Transcriptomic analysis of these tumours revealed that insulin signalling and inflammation were the possible mechanisms that underly the prognostic effect of obesity on ER+ BC. GSEA showed that protein kinase B (AKT) target genes, as well as genes involved in glucose metabolism, in the generation of precursors of metabolites and energy, as well as in the epithelial–mesenchymal transition and metastasis, were upregulated in patients with obesity [81]. To explore the causal relationship between obesity and tumour progression in ER+ BC, Fuentes-Mattei et al. generated oncogene-induced BC obese mouse and lean mouse models [81]. Transcriptomic analyses of these tumours suggested that obesity was associated with tumour metastasis, invasion, inflammation and cell death resistance, which were mediated by oestrogen signalling, hyperinsulinemia, IGF-1 and adipokine secretion [81]. The role of peripheral inflammation and BMI in ER+ breast tumours was also supported by a study of 216 BC patients by Madeddu et al. In this study, leptin levels in ER+ patients were significantly higher compared to those in ER− patients. Multivariate regression analysis revealed that BMI, leptin, IL-6 and reactive oxygen species were predictive factors for tumour size, lymph node stage and metastasis status in ER+ patients [82]. These findings indicate that an imbalance in the secretion of adipokines may play a role in the link between obesity and ER+ BC. This imbalance may result in a pro-inflammatory environment that promotes tumour progression and metastasis.

Quigley et al. compared gene expression profiles in 195 breast tumours of all subtypes to gene expression in matched adjacent normal tissue and tissue from women who underwent mammoplastic reduction surgery [83]. The expression of cytokines associated with acute inflammatory response such as IL-1B, TNF-α and suppressor of cytokine signalling 3 were significantly higher in the normal tissue adjacent to the breast tumour compared with mammoplastic reduction samples from healthy donors. BMI was also significantly correlated with the expression of leptin and macrophage scavenger receptor, and with macrophage pathway expression levels in adjacent normal tissue but not in tumours. The change in macrophage pathways was inversely correlated with the change in oestrogen receptor-a expression in ER+ but not ER− patients. ER− tumours highly expressed macrophage pathways compared to matched adjacent normal tissue. On the contrary, there was an inverse correlation between macrophage pathway expression and oestrogen receptor-a expression in the tumour of ER+ cancers compared with adjacent normal tissue. The effect of the oestrogen signalling pathway on inflammation in breast tumours was investigated by Qureshi et al., who showed that oestrone, which is upregulated in post-menopausal women, promotes NF-kB-mediated inflammation which results in the increase of tumour-initiating stem cells and ER+ cancer initiation and progression, as well as poor outcomes [84]. Thus, high BMI is associated with inflamed normal tissue adjacent to the tumour, which induces aggressive phenotypes that are mediated by oestrogen signalling in ER+ breast tumours.

In summary, adiposity and, by extension, subclinical inflammation, are reported to be associated with changes in the immune-tumour microenvironment in BC that can be subtype-specific. Obesity has also been correlated with aggressive clinical and molecular phenotype and enriched inflammation pathways.

## 10. Adiposity and Response to Cancer Immunotherapy in Breast Cancer

Recent advances in cancer immunotherapy with ICPIs have significantly improved clinical outcomes in patients with tumour types that were previously difficult to treat [85,86,87]. Nevertheless, only a limited proportion of patients demonstrate durable responses to immunotherapy [85,86,87]. This can be potentially explained by several parameters that affect immunotherapy responses such as the presence of TILs, the absence of immune checkpoints, systemic inflammation, hypoxic TME and low tumour mutational burden [88]. Breast tumours are considered immunologically quiescent compared to other tumour types. The heterogeneity in immunological profiles and mutational burden among the different subtypes of BC partially explains the differential responses to ICPIs. For instance, TNBC, especially the basal subtype, as well as HER2+ breast carcinomas, are associated with higher mutational burden and higher frequency of TILs compared to the ER+ tumours [89,90]. A higher rate of TILs has been associated with improved clinical outcomes in patients with HER2+ and TNBC following neoadjuvant chemotherapy [89]. TILs can also identify a subset of patients with stage I TNBC who may not require adjuvant chemotherapy [91]. Chemotherapy may promote the release of tumour neoantigens and consequently tumour-specific immune responses. ICPIs in combination with chemotherapy have been evaluated in patients with TNBC where pembrolizumab and atezolizumab have been approved in neoadjuvant and metastatic settings [92,93]. Currently, PD-L1 expressed by the tumour is used as a biomarker to select patients with metastatic BC who are most likely to benefit from ICPIs. However, PD-L1 is not used in the primary TNBC setting. Despite the positive results, it is not clear who can benefit from immunotherapy in this group of patients.

Randomised clinical trials investigating the PD-1/PD-L1 blockade, either as monotherapy or in combination with chemotherapy or targeted agents, in patients with metastatic BC, demonstrated objective responses and median OS ranging from 0% to 52% and from 8.1 to 17.2 months, respectively [89,92,94]. Higher responses were confined to patients with triple negative, HER2+, PD-L1+, BRCA1/2 deficient and treatment-naïve subgroups of BC patients, underlying the fact that person stratification is important for the optimisation of immune responses in BC patients [89,92,94]. Nevertheless, none of these published RCTs stratified the patients with BC by body composition parameters, indicating the need to explore the potential impact of metabolic parameters on immune responses, disease progression and treatment outcomes in RCTs.

Currently, the evidence on body composition and immunotherapy responses is derived from other types of solid tumours, and data on BC are lacking. A systematic review and meta-analysis evaluated the association between BMI and the efficacy of ICPIs in 5279 patients with cancer (renal cell carcinoma, malignant melanoma and lung cancer) from 13 retrospective cohorts [95]. This study showed improved clinical outcomes in patients with high BMI who were treated with ICPIs compared to patients with healthy BMI with no statistically significant difference in the frequency of immune-related adverse events [95]. This observation is supported by Maslov et al., who demonstrated that high BMI in patients with metastatic cancer, across 20 different tumour types, is associated with a 48% lower risk of disease progression or death compared to patients with healthy BMI [96]. In addition, in a cohort of 250 patients diagnosed with several types of cancers who were treated with ICPIs, individuals with a BMI ≥ 30 had significantly better progression-free survival and OS compared to patients without obesity [31]. Comparable results were shown in a multi-cohort analysis that included two cohorts of patients with advanced melanoma who received ICPIs [97]. In the first cohort, 207 patients received ipilimumab plus dacarbazine, whereas in the second cohort, 329 patients were treated with either pembrolizumab, nivolumab or atezolizumab. The pooled analysis showed that patients with a BMI ≥ 30 demonstrated longer progression-free survival and OS compared to non-obese individuals. These observations may be explained by the elevated levels of systemic inflammation in patients with high BMI. This is supported by a small cohort of 26 healthy human volunteers, in which significantly elevated levels of serum leptin were observed in individuals with obesity compared to those without, which was also associated with higher PD-1 expression on CD8+ T-cells [31].

Tumours evolve multiple mechanisms of immune escape and new therapeutic strategies to prevent this are of particular significance [98]. Western diet and obesity can promote systemic chronic inflammation with increased proinflammatory cytokines such as leptin [60]. Leptin is an established pro-inflammatory cytokine that has effects both on innate and adaptive immunity. Immune cells express leptin receptors and leptin promotes proliferation of naïve T-cells, induces Th-1 responses and restores the function of impaired T-cells [99]. Leptin also contributes to the activation of monocytes and macrophages via the upregulation of proinflammatory cytokines, promotes survival of dendritic cells and neutrophil chemotaxis, and acts as a negative regulator for the proliferation of Tregs via the activation of the mTOR pathway [99]. Wang et al. investigated the effect of obesity and leptin levels on T-cell responses in multiple species and tumour models [31]. Upregulation of PD-1 in CD8+ T-cells, together with reduced proliferative capacity, INF-γ and TNFα production, were observed in diet-induced obese mice compared to controls [31]. Similar findings were reported in non-human primates and healthy human donors, which were stratified by body weight and BMI, respectively [31]. In addition, obesity promotes tumour growth and T-cell exhaustion with PD-1 upregulation in tumour-infiltrating CD8+ T-cells in both B16F0 melanoma and 4T1 BC murine models, which were shown to be partly mediated via leptin [31]. Zhang et al. showed that leptin in breast adipose tissue downregulates the effector functions of CD8+ T-cells by activating the STAT3-Fatty Acid Oxidation (FAO) pathway and by inhibiting glycolysis and INF-γ secretion [100]. In addition, PD-1 increases FAO and inhibits IFN-γ and glycolysis via STAT3 activation in the tumour-infiltrating CD8+ T-cells [100]. The PD-1 signalling pathway reprograms T-cells metabolically by suppressing AKT and mTOR signalling, which also leads to FAO rather than glycolysis [101]. The anti-PD-1 blockade resulted in higher responses, significant reduction in the tumour burden and improvement in survival in obese compared to lean B16 tumour-bearing mice [31]. Moreover, higher CD8+ T-cell tumour infiltration and reduction in the metastatic burden in B16-bearing mice was reported [31]. These results are supported by Dyck et al., who showed fewer tumour-infiltrating lymphocytes that were characterised by decreased proliferation and cytokine secretion in high-fat-diet compared to standard-diet MC38 and B16-F10 tumour-bearing mice [102]. Overall, leptin is associated with a STAT3-mediated metabolic reprogramming of CD8+ T-cells, which compromises their anti-tumour properties.

Collectively, obesity and systemic inflammation are associated with upregulation of immune checkpoints on T-cells, which is partially mediated via the immunomodulatory effects of leptin in pre-clinical tumour models and clinical studies. In chronic inflammatory conditions such as obesity, T-cells become exhausted due to the persistent antigenic exposure, which leads to the expression of inhibitory receptors and metabolic reprogramming of these immune cell populations [103,104]. In addition, tumour-associated macrophages can directly suppress T-cell responses via overexpression of inhibitor receptors and by interfering in signalling pathways that are involved in inflammation, metabolism or proliferation and hypoxia [105,106]. This has been paradoxically associated with improved responses to ICPIs in several cancers. Currently, there is lack of evidence about the role of body composition and metabolic parameters in immune responses in BC. Hence, further research is needed in this field that can be potentially used to optimise patient stratification and personalisation of BC treatment.

## 11. Metabolic Interventions in Immunometabolic Reprograming in Breast Cancer: The Paradigm of Metformin

Metformin, a dimethyl biguanide antidiabetic drug with pleiotropic effects, has demonstrated anti-tumourigenic properties both in pre-clinical and clinical studies [13]. In vitro and in vivo pre-clinical models of endometrial cancer showed that metformin reverses obesity-induced tumour aggressiveness by downregulating lipid and protein biosynthesis in obese compared to lean mice [107]. In addition, in a phase II randomised clinical trial (RCT), premenopausal women with obesity or overweight and characteristics of metabolic syndrome without history of BC were treated with metformin or placebo [14]. After 12 months of treatment, metformin was associated with a significant reduction in waist circumference and waist-to-hip ratio, as well as a decrease in markers of systemic inflammation such as serum leptin and the leptin-to-adiponectin ratio and neutrophil-to-lymphocyte ratio, compared to the placebo group [14]. Together, these findings suggest that metformin may have anti-tumour properties via the modulation of obesity-mediated systemic inflammation and tumour metabolism.

Previous studies demonstrated that metformin exerts pleiotropic anti-tumour effects that involve tumour metabolism, cell cycle, DNA repair mechanism, reactive oxygen species, angiogenesis and inflammatory pathways as shown in pre-clinical studies. In addition, metformin modulates the density of immune cell infiltrates in human and murine tumours [13,108,109]. Previous studies also showed that metformin polarised M2-like macrophages to the M1-like phenotype within the tumour microenvironment, which led to the recruitment of CD8+ T-cells into the tumour and reduction in immunosuppressive infiltration of myeloid-derived suppressor cells and regulatory T-cells, which may be mediated by the secretion of proinflammatory cytokines [110,111,112,113]. Specifically, metformin is associated with increased CD68+ and F4/80+ and decreased CD11c+ and CD206+ macrophage densities in murine models [114,115]. In vitro and in vivo experimental models showed that metformin was correlated with reduced secretion of anti-inflammatory and increased production of pro-inflammatory cytokines by macrophages [116,117]. Regarding T-cells, metformin was associated with higher CD8+ T-cell density and reduced CD4+Foxp3+ regulatory T-cells, in vivo [118,119]. Interestingly, there was an increase in the density of regulatory T-cells derived from B16F10 murine melanoma tumours [120]. In addition, tumour-infiltrating lymphocytes were associated with enhanced production of proinflammatory cytokines, granzyme B and perforin after the administration of metformin both in vitro and in mice [120,121]. Furthermore, tumours downregulated the expression of PD-L1 and restored the MHC-1 expression [118,122]. These findings suggest that metabolic reprogramming of macrophages with the use of metformin may repolarise tumour-associated macrophages towards an anti-M1-like phenotype that could then enhance CD8+ T-cell effector function and suppress the recruitment of regulatory T-cells. Thus, the combination of metformin and ICPIs may further enhance immune responses. Nevertheless, further research is required to optimise the identification of subgroups of BC patients that may benefit from metformin.

Pre-clinical studies evaluated the efficacy of metformin in combination with ICPIs in several solid tumours including BC. In 4T1 murine BC experimental models, the use of a nanodrug that contained metformin and the anticancer agent SN38 (7-ethyl-10-hydroxycamptothecin) resulted in improved outcomes compared to metformin or ICPIs alone, an effect that was mediated by the downregulation of PD-L1 expressed by the tumour cells [123]. Cha et al. demonstrated that metformin-activated AMPK directly binds to and phosphorylates PD-L1, which induces abnormal glycosylation, leading to endoplasmic reticulum-associated protein degradation [118]. In addition, in MYC-overexpressed breast tumour models, metformin combined with anti-apoptotic B-cell lymphoma-2 inhibitors, navitoclax or venetoclax, led to tumour growth inhibition, improved clinical outcomes and tumour infiltration by immune cells [124]. In another study, where 4T1 murine breast tumours were treated with metformin-loaded mannose-modified macrophage-derived microparticles (Man-MPs), it was reported that metformin polarised M2-like macrophages to the M1-like phenotype within the TME [110]. This resulted in the recruitment of CD8+ T-cells into tumour tissues and reduction in myeloid-derived suppressor cells and regulatory T-cells. Similarly, in experimental lung cancer and malignant melanoma murine models, the combination of metformin and anti-PD-1 ICPIs was associated with CD8+ T-cell tumour infiltration, reduction in myeloid-derived suppressive cells, inhibition of tumour growth and improved outcomes [111,112,113,125,126]. Hence, pre-clinical studies that evaluated the efficacy of metformin in combination with ICPI or other novel drugs in several solid tumours including BC showed promising results. Nonetheless, further experimental models need to be developed to evaluate immunometabolic reprogramming in BC, which will improve personalisation of treatment and enable us to support clinical randomised controlled trials to investigate metabolic immune reprogramming with and without immunotherapy in patients with primary BC.

Despite promising pre-clinical data, randomised clinical trials (RCTs) evaluating the impact of metformin on clinical outcomes did not show statistically significant results. Specifically, a recent phase III, double-blind RCT that included high-risk non-metastatic and non-diabetic BC patients showed no difference in disease-free survival and OS between patients randomised to adjuvant metformin or placebo in the ER/PR+ group [127]. However, in the same study, exploratory analysis in the HER2+ group revealed a statistically significant improvement in disease-free survival and OS in the metformin arm [127]. Similarly, a systematic review and metanalysis that included three cohort studies of diabetic patients with stage I-III BC, who received adjuvant metformin, showed no difference in recurrence-free, overall, and cancer-specific survival between metformin and controls [128]. In addition, three phase II RCTs with non-diabetic patients diagnosed with metastatic BC failed to show any benefit in the metformin arm compared to the control group [129,130,131]. In contrast, in a neoadjuvant setting there was a non-statistically significant trend of association between metformin and improved objective response rate in non-diabetic patients with locally advanced disease [132,133]. Furthermore, in a retrospective cohort of patients with metastatic malignant melanoma treated with ICPIs with or without metformin, patients who received combination treatment demonstrated a trend of improved clinical outcomes, an association that was non-statistically significant likely due to the small sample size [134].

Although the clinical studies showed negative results, they were characterised by certain limitations. First, except for those by Goodwin et al. and Lega et al., the studies were characterised by a small sample size, which may have introduced random error. Secondly, none of the studies focused on a specific BC subtype. Hence, tumour heterogeneity might have confounded any possible clinical benefit related to metformin treatment. Furthermore, these studies did not stratify patients by metabolic or inflammatory parameters such as serum leptin or CRP, which may help optimise the identification of patients who may have clinical responses to metformin. Other possible confounders include local adiposity and inflammation such as the presence of CLSs, and body composition such as Fat Mass Index (FMI) and BMI, which may affect response to treatment. Moreover, the higher pathological response in patients who received neoadjuvant metformin may indicate that metformin may have a direct tumour effect and that it may be best given in a neoadjuvant setting. However, these studies had a small sample size and might have been underpowered.

Two key preoperative window-of-opportunity clinical trials investigated the mechanism of action of metformin in non-diabetic patients with early primary BC [135,136]. The first trial by Hadad et al. recruited 47 non-diabetic patients who were randomly allocated to a two-week course of metformin or no medication [135,137]. Immunohistochemistry for metabolic and cell proliferation markers was performed on formalin-fixed paraffin-embedded core biopsies that were taken at baseline and post-treatment [135]. RNA microarray analysis in metformin and control samples was also carried out [137]. In patients treated with metformin there was an increase in the expression of tumour pAMPK, reduction in pAkt and suppression of insulin responses, as well as a significant decrease in Ki67 and cleaved caspase-3, suggesting that metformin may exert a cytostatic anti-tumour effect [135]. Ingenuity pathway analysis demonstrated that metformin was associated with pathways involved in TNFR1, cell cycle regulation and metabolism [137]. The second window-of-opportunity study by Lord et al. integrated dynamic PET imaging, metabolomics and transcriptomics pre- and post-metformin in a single arm cohort of 41 non-diabetic patients with primary BC. This study demonstrated evidence that metformin is associated with low levels of mitochondrial metabolites, with activation of multiple mitochondrial metabolic pathways, and increase in 18-FDG flux in tumours [136]. Moreover, two distinct metabolic responses were identified, the glycolytic and oxidative phosphorylation responders with the latter conferring resistance to metformin [136]. Overall, metformin appears to have an anti-tumour effect that is mediated by metabolic and cell cycle pathways.

In summary, metformin modulates the tumour microenvironment via the enhancement of anti-tumour immune responses which may be partially mediated by the secretion of cytokines by both T-cells and macrophages. Although RCTs showed no evidence of association between metformin and improved clinical outcomes in patients with BC, pre-clinical studies showed that the combination of metformin and ICPIs in breast cancer, lung cancer and malignant melanoma murine models is correlated with enhanced tumour-infiltrating lymphocytes and improved outcomes. This discrepancy in the findings between clinical and pre-clinical studies may be explained by biological or methodological reasons. For instance, all tumour subtypes were included, indicating that tumour heterogeneity might have masked any possible associations. In addition, these studies did not stratify patients by metabolic or inflammatory parameters, which may help optimise the identification of patients who may respond to metformin. Moreover, the small sample size and confounding factors such as anthropometric, clinical and metabolic parameters might not have been considered, whilst pre-clinical studies were characterised by a well-defined population and controlled conditions that minimised confounders and bias. There are a limited number of human studies that studied small cell lung cancer and malignant melanoma and that showed weak evidence of better responses and survival outcomes, respectively [134,138]. However, these studies are characterised by their retrospective nature and small sample sizes. Nevertheless, the immunomodulatory properties of metformin and potential underlying mechanism in patients with BC require further investigation.

## 12. Future Directions

Metabolic interventions may play a direct role in the metabolic reprogramming of T-cell and macrophages populations that are dysregulated by systemic inflammation. Experimental models need to be developed to evaluate immunometabolic reprogramming in breast cancer. Multiomic evaluation of the effect of metformin and ICPI combination on the TME in breast tumour models will help us understand the underlying biology and immunomodulatory effects of the ICPI and metformin combination. These findings will enable us to support a window-of-opportunity RCT to investigate metabolic immune reprogramming with and without immunotherapy in breast cancer. In addition, the use of markers of systemic inflammation, metabolic and body composition parameters, and the presence of CLSs, as a potential biomarker, may help stratify the patients and improve personalisation of treatment, which can inform the design of future RCTs evaluating novel immunometabolic treatment combinations.

## 13. Conclusions

There is mounting evidence that chronic inflammation in adipose tissue is associated with the promotion of protumourigenic macrophages and exhausted T-cell phenotypes that are linked to resistance to anti-cancer therapy [60,139] (Figure 2). Although the underlying biology is complex, a possible mechanism is that chronic inflammation enhances metabolic competition between tumour and immune cells for nutrients in favour of the tumour, which leads to metabolic dysregulation of macrophages and T-cells [140]. In obesity, systemic inflammation is initiated, promoted and maintained partly via the metabolic reprogramming of adipose tissue macrophages [51,53,54,55,59,65,141]. Obesity is also associated with the upregulation of immune checkpoints on T-cells, which has been correlated to improved responses to ICPIs in several cancers, suggesting that body composition can be used to optimise patient personalisation of treatment. Metabolic interventions such as metformin targeting this metabolic dysregulation may potentially restore anti-cancer function of the immune cells and improve clinical outcomes in patients with BC. This effect may be mediated by the modulation of obesity-mediated systemic inflammation, tumour metabolism and tumour-immune microenvironment. The combination of metformin and ICPIs in murine models is associated with enhanced tumour-infiltrating lymphocytes and improved outcomes. However, prospective studies considering body composition and metabolic parameters are required to investigate metabolic immune reprogramming with and without immunotherapy in patients with BC.

## Figures and Tables

**Figure 1 cancers-15-02440-f001:**
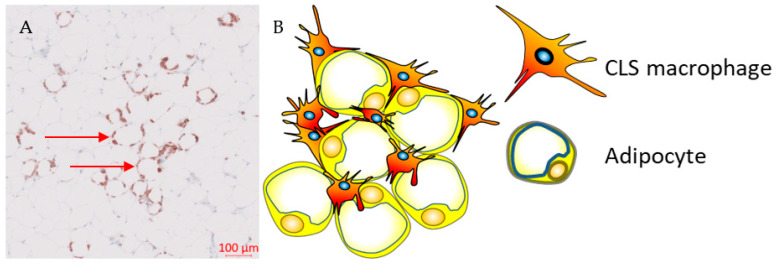
Crown-like structures. (**A**) Representative immunohistochemistry image from the Southampton BEGIN (Investigating outcomes from breast cancer: Correlating genetic, immunological, and nutritional predictors) cohort showing CD68+ crown-like structures within the adipose tissue adjacent to a human breast tumour; (**B**) hypertrophic adipocytes surrounded by macrophages forming crown-like structures.

**Figure 2 cancers-15-02440-f002:**
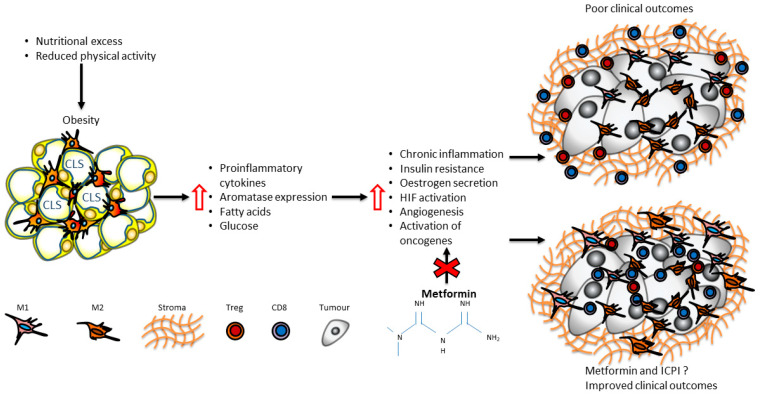
Proposed role of obesity and metformin in breast cancer immune-tumour microenvironment. Obesity may be associated with inflammatory signatures and an aggressive molecular phenotype that may promote cellular proliferation, differentiation and tumour growth, as well as suppressive T-cell phenotypes and signatures that may lead to immune escape.

**Table 1 cancers-15-02440-t001:** Obesity is associated with systemic immunometabolic changes.

Immune Changes	Metabolic Changes
Upregulation of proinflammatory signalling pathways [24]	Increased insulin and IGF levels [28]
Increased immune cell infiltration [29]	Insulin resistance [20,27]
Upregulation of WNT signalling [30]	Elevates leptin levels [31]
Increased synthesis of arachidonic acid and PGE2 [32]	Increases oestrogen and androgen levels [33,34]
Downregulation of response to antigen and mitogen stimulation [35,36]	Anti-apoptotic, promotes stemness [37]

**Table 2 cancers-15-02440-t002:** Clinical outcomes of retrospective cohorts that evaluated the prognostic role of CLSs in breast cancer.

Study	Sample Size	CLS− (n)	CLS+ (n)	CLS Marker	RFS, CLS+ vs. CLS−	OS, CLS+ vs. CLS−
Iyengar N (cohort 2) (2016) [58]	127	75	52	CD68	**1.83** (1.07–3.13) ^a,c^	not reported
Koru-Sengul T ^f^ (2016) [56]	134	NR	NR	CD40	5.87 (0.73–47.23) ^a,c^	**13.59** (1.56–118.16) ^a,c^
Koru-Sengul T ^f^ (2016) [56]	134	NR	NR	CD163	2.21 (0.65–7.59) ^a,c^	2.42 (0.54–10.89) ^a,c^
Koru-Sengul T ^f^ (2016) [56]	134	NR	NR	CD206	1.17 (0.09–15.35) ^a,c^	0.74 (0.04–15.55) ^a,c^
Cha YJ (2018) ^g^ [55]	140	122	18	CD163	105 (94–116) vs. 124 (118–131) ^b,d^	105 (94–116) vs. 130 (124–136) ^b,d^
Cha YJ (2018) ^g^ [55]	140	115	25	CD68	106 (97–114) vs. 124 (117–131) ^b,d^	106 (99–114) vs. 130 (124–136) ^b,d^
Cha YJ (2018) ^g^ [55]	56	49	7	CD68	76 (56–96) vs. 120 (108–132) ^b,d,e^	79 (63–96) vs. 125 (114–136) ^b,d,e^
Maliniak M (2020) [59]	319	223	96	CD68	1.05 (0.64–1.72) ^a,c^	1.02 (0.55–1.87) ^a,c^
Birts C (2022) [45]	117	47	61	CD32B	**4.2** (1.01–17.4) ^a,c^	not reported

Abbreviations: CLS, Crown-like structures; NR, not reported; RFS, Recurrence Free Survival; OS, Overall Survival; ^a^ HR (95%CI); ^b^ Mean, month (95% CI); ^c^ Multivariate analysis; ^d^ Univariate analysis; ^e^ Node-positive patients; ^f,g^ same cohort; bold, statistically significant.

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
