# Peer review of "Obesity Is Associated with Immunometabolic Changes in Adipose Tissue That May Drive Treatment Resistance in Breast Cancer: Immune-Metabolic Reprogramming and Novel Therapeutic Strategies"

_cancers, 2023, doi:10.3390/cancers15092440_

Round 1

Reviewer 1 Report

The review by Savva et al is very well written and comprehensive. I only have a few comments that should be easy to address.

Line 47: “in both sexes” makes it sound like BC is the most common disease in both men and women, which is not the case. Please clarify

Line 56 and 58: references 3 and 4 do not seem to be correctly placed. Reference 3 is about obesity prevalence but the sentence is about obesity and BC survival. And vice versa for reference 4

Line 90: citation 16 is a review paper but original research papers should be cited. Overall, this section over-uses review papers for citations. Please refer to the original research as much as possible (eg. citation 21, line 99).

Table 1: citations should be added

Table 2: this table is not positioned correctly, it should come after the first mention. This table is quite confusing with respect to the RFS and OS columns because each line is a different measurement. The high number of footnotes makes the table difficult to understand. The formatting of the footnotes should match the formatting in the table (ie superscripts for both). Is there any way to improve the layout to reduce the need for so many footnotes?

The citations should be added so that these references can be easily found within the reference list.

Figure 1: is this an original image in panel A? If so, can a better quality image be shown? Regardless, please indicate the source in the figure legend.

Line 133: immunity should be immune

Lines 169-172: Please expand on mechanism behind the observation with CD32B.

Lines    ~350-383: The authors argue, convincingly, that obese patients respond better to ICPI because obesity leads to an increase in leptin which leads to an increase in PD-L1 expression. But in lines 355-356, state that leptin is associated with resistance to trastuzumab. This needs clarified.

Reviewer 2 Report

Manuscript ID # cancers-2273577

Manuscript Title:  Obesity is associated with immunometabolic changes in adipose tissue that may drive treatment resistance in breast cancer: immune-metabolic reprogramming and novel therapeutic strategies.

Authors: Constantinos Savva, Ellen Copson, Peter WM Johnson, Ramsey I Cutress, Stephen A Beers

Corresponding Author(s): Ramsey I Cutress, Stephen A Beers

General Comments

Savva C, et al., in their comprehensive review article entitled, “Obesity is associated with immunometabolic changes in adipose tissue that may drive treatment resistance in breast cancer: immune-metabolic reprogramming and novel therapeutic strategies” have discussed the link between obesity-related immunometabolic changes and BC incidence and progression and therapeutic resistance. The theme is important and timely, and the article is well structured. 

Specific Comments

1)       The introduction starts with the sentence, "BC is the most common type of malignancy worldwide in both sexes". It is true that BC has the highest incidence rate among all cancer types when both sexes are considered. However, this is true for females, although cases of BC have been reported in males. Among males, the most prevalent is lung cancer. For example, if this manuscript was on prostate cancer, it would be like saying, "prostate cancer is the second most common type of malignancy worldwide in both sexes" while this is applicable only to males. Hence re-phrase this sentence to bring the apt meaning.

2)       There seems to be an obvious link between obesity and TNBCs. TNBCs are more aggressive and quite often develop resistance to therapeutic intervention. However, the authors have given little information in this regard. Adding a separate section to address this would be helpful.

3)       Metformin suddenly appears from page 9, line 398 onwards, while there is no mention of it in the abstract or introduction. Why? What was the reason/basis for the authors to choose metformin as the drug of choice when considering obesity and BCs? Justify and provide this information in the text, including brief mentions in the abstract and introductions.

4)        Additional 1 or 2 diagrammatic representations or pathway illustrations are recommended (for instance for the role of metformin as an anti-cancer agent) to make the manuscript more readable and easily understandable.

5)       Provide a paragraph or two on future directions or perspectives.
